# Targeting Aquaporins in Novel Therapies for Male and Female Breast and Reproductive Cancers

**DOI:** 10.3390/cells10020215

**Published:** 2021-01-22

**Authors:** Sidra Khan, Carmela Ricciardelli, Andrea J. Yool

**Affiliations:** 1Discipline of Physiology, Adelaide Medical School, The University of Adelaide, Adelaide, SA 5005, Australia; sidra.khan@adelaide.edu.au; 2Discipline of Obstetrics and Gynaecology, Robinson Research Institute, Adelaide Medical School, The University of Adelaide, Adelaide, SA 5005, Australia; carmela.ricciardelli@adelaide.edu.au

**Keywords:** aquaporins, AQPs, reproductive cancer, metastasis

## Abstract

Aquaporins are membrane channels in the broad family of major intrinsic proteins (MIPs), with 13 classes showing tissue-specific distributions in humans. As key physiological modulators of water and solute homeostasis, mutations, and dysfunctions involving aquaporins have been associated with pathologies in all major organs. Increases in aquaporin expression are associated with greater severity of many cancers, particularly in augmenting motility and invasiveness for example in colon cancers and glioblastoma. However, potential roles of altered aquaporin (AQP) function in reproductive cancers have been understudied to date. Published work reviewed here shows distinct classes aquaporin have differential roles in mediating cancer metastasis, angiogenesis, and resistance to apoptosis. Known mechanisms of action of AQPs in other tissues are proving relevant to understanding reproductive cancers. Emerging patterns show AQPs 1, 3, and 5 in particular are highly expressed in breast, endometrial, and ovarian cancers, consistent with their gene regulation by estrogen response elements, and AQPs 3 and 9 in particular are linked with prostate cancer. Continuing work is defining avenues for pharmacological targeting of aquaporins as potential therapies to reduce female and male reproductive cancer cell growth and invasiveness.

## 1. Introduction

Aquaporins (AQPs) in the major intrinsic protein (MIP) family, found in all forms of life from bacteria to mammals, have attracted interest as therapeutic targets [1,2,3]. As summarized in Table 1, the first AQP channel was isolated from red blood cells as a 28 kDa protein called CHIP28, which mediated osmotic water transport [4,5], and was named Aquaporin-1. At least 17 mammalian aquaporins have been identified to date, with AQP0-12 found in higher orders including human, and AQP13-16 described in older lineages [6,7]. The aquaporin family traditionally has been classified into three general groups on the basis of amino acid sequence homologies and permeability characteristics, usually measured in heterologous expression systems such as *Xenopus* oocytes, a method pioneered by Preston and colleagues [5]. Oocytes injected with copy RNA encoding an AQP acquire high permeabilities not seen in native oocytes to water, glycerol, urea, ions, and/or other solutes, depending on the AQP subtype expressed and the presence of appropriate signaling environments [8,9]. In higher mammals, classical aquaporins known for permeability to water include AQP0, AQP1, AQP2, AQP4, AQP5, AQP6, and AQP8. Aqua-glyceroporins also permeable to small uncharged solutes such as glycerol include AQP3, AQP7, AQP9, and AQP10. A third group called subcellular aquaporins consists of AQP11 and AQP12, which show low sequence homology with other aquaporins and have been difficult to evaluate due to predominant localization in intracellular organelles [10]. AQP11, when successfully expressed in human adipocytes, was found be permeable to water and glycerol [11]; reconstituted purified AQP12 shows water flux [12] but channel activity in cells remains to be characterized.

The classes of aquaporins in the human body show patterns of tissue expression that correspond with organ functions, accomplished at the cellular level by aquaporin-mediated fluxes of fluids and solutes [28] including for example kidney filtration and water reabsorption [29,30], glycerol transport and fat metabolism in liver and adipose tissues [31,32], cell migration [33,34], cerebral spinal fluid production [35,36,37], vision [38,39], and skin hydration and wound healing [40]. Changes in aquaporin gene expression have been linked to human diseases such as congenital bilateral cataracts [41], obesity [31], nephrogenic diabetes insipidus [42,43], Sjögren’s syndrome [44,45], cerebral edema [46,47], pulmonary edema [48], and cancer [9,49,50], to mention a few. Traditional distinctions based on solute selectivity are becoming blurred as more substrates are being added steadily to the growing repertoires of AQP classes [51]. The full spectra of solutes permeable through AQP channels remain to be defined. In addition to water, AQP1 allows transport of CO_2_ [52], H_2_O_2_ [53,54], NO [55], and NH_3_ [56]. Several classes of mammalian AQPs function as ion channels. AQP0, a major component of lens fiber, has ion channel activity [57,58]. AQP1 conducts monovalent cations after activation by intracellular cyclic GMP [59]. AQP6 is permeable to anions such as nitrate and chloride at low pH [60,61]. Human AQP8 is permeable to ammonium analogues [62]. More ion-conducting aquaporins are likely to be discovered. Reconsideration of the current classification system may be merited.

## 2. Structure and Function of Aquaporins

Aquaporins exist as tetramers; each monomer has six transmembrane helices connected by loops A to E, with intracellular carboxy and amino terminal domains (Figure 1A). Hydrophobic segments of loops B and E contain the signature NPA motifs (N = asparagine, P = proline, and A = alanine) that fold together within each monomer to form a narrow constriction with a diameter less than 3 A° in AQP1, which with adjacent aromatic and arginine residues creates a selectivity filter allowing single file movement of water [63,64]. In AQP1, the tetrameric central pore (Figure 1B) is thought to be responsible for gated ion currents [65,66].

## 3. Roles of Aquaporins in Cancer Metastasis, Angiogenesis, and Apoptosis

### 3.1. AQPs in Cancer Metastasis

Cancer metastasis, the spread of cancer cells from tumor sites of origin to distant sites, is responsible for 90% of cancer-related deaths [67]. Different classes of aquaporins have been implicated in processes of cancer metastasis, depending on cancer type [8], and are thought to contribute to the cell motility in part by facilitating cell volume changes in response to osmotic gradients [68]. Depolymerization of actin filaments at leading edge of tumor cells has been proposed to increase local intracellular osmotic pressure, causing an influx of water and solutes through aquaporin channels polarized at the leading edges of tumor cells that drives protrusions and lamellipodial formation, and thus migration [68,69]. Genetic knockdown of AQP1 in melanoma and colon cancer cell lines was associated with reorganization of actin cytoskeleton, and decreased cell migration [69,70]. Pharmacological inhibition of aquaporins slowed cancer cell migration; both ion conduction through the central pore of AQP1 and water flux through intrasubunit pores were necessary for maximal cell migration in an in vitro model of colon cancer [71,72].

Current cancer treatments are focused on inhibiting cell proliferation and arresting cell cycle, controlling angiogenesis, and modifying stromal microenvironments, but tools for controlling metastasis remain elusive. Multiple steps in metastasis involve dissociation of cells from the primary tumor, invasion through extracellular matrix, and intravasation into blood and lymphatic vessels to spread to a new location [67]. In addition to motility, multiple steps appear to be influenced by AQP expression patterns. Increased levels of AQP1 expression in lung carcinoma were associated with downregulation of epithelial (E-) cadherin [73]. E-cadherin is a glycoprotein that mediates tight junctions between cells and links to cytoskeletal elements within cells; reduced levels are a hallmark of tumor metastasis [74]. Conversely, genetic knockdown of AQP4 in breast cancer cell lines T47D and MCF7 correlated with increased E-cadherin levels, with reduced migration and invasiveness [75]. The strikingly different roles of different classes of AQPs suggests that precise targeting of subtypes could allow customized interventions tailored to cancer type.

### 3.2. AQPs in Angiogenesis

Rapidly dividing cancer cells have a high metabolic demand and require reliable access to oxygen and nutrients for survival; tumors cannot grow beyond few millimeters without a continuous supply of nutrients [76]. Inadequate blood flow and poor clearance of metabolic waste from tumor masses result in hypoxia, triggering transcription of growth factors such as vascular endothelial growth factor (VEGF), transforming growth factor-β, and platelet-derived growth factor hypoxia induced factor-1 (HIF-1) that in turn induce angiogenesis, the formation of new blood vessels [77]. Aquaporins appear to be necessary for the induction and establishment of angiogenesis. Levels of AQP1 expressed in endothelial cells of peripheral blood vessels correlated with release of VEGF, resulting in angiogenesis and increased tumor growth in endometrial carcinoma [78]. Hypoxia increased expression of AQP1 and AQP4 in rat model of tissue ischemia [79,80]. Reduced levels of AQP1 in vitro by small interfering RNA [70] and in vivo by genetic knockdown led to decreased angiogenesis and corresponding reductions in tumor growth and invasiveness [81,82]. AQP1-null mice showed a reduced vascularity and substantial necrosis of introduced tumors as compared to tumors hosted in wild type mice [82]. Pharmacological inhibition of AQP1 water channel activity in a colon cancer cell line impaired endothelial tube formation [50]. Pharmacological targeting of AQP1 to inhibit angiogenesis could provide a useful adjunct therapy.

### 3.3. AQPs in Apoptosis

Apoptosis is a pattern of programmed cell death [83] essential for normal tissue development, repair, and homeostatic control throughout life. Certain aquaporins (AQP4, -8, and -9), in parallel with ion channels and pumps, have been proposed to facilitate the early stage of cell shrinkage initiating apoptosis [84]. Adaptive responses resulting in resistance to apoptosis can involve downregulation of these AQPs [85]. Genetic knockdown of AQP1 in rat granulosa and Chinese hamster ovary cells was associated with protection from cell shrinkage and apoptosis [86]. AQP3, located in plasma membranes of human prostate tissue and benign tumors, was found to be internalized in prostate cancer cells, again consistent with a reduction in AQP functionality promoting resistance to apoptosis [87]. In contrast, other cases involving AQP1 and AQP5 have yielded opposite outcomes, in that the presence of the functional channel was protective, and loss of function promoted cell death. For example, apoptosis was induced by pharmacological inhibition of AQP1 in colon cancer cells [50], and by siRNA knockdown of AQP1 in esophageal cancer cells [88]. AQP5 overexpression in esophageal cancer cells was associated with resistance to apoptosis [89]. More research is needed to clarify the pro- versus anti-apoptotic roles of different classes of aquaporins by cancer type and conditions. Therapeutic strategies for reproductive cancers ultimately could harness agents that increase or decrease levels of activity of endogenously expressed classes of aquaporins in females (Figure 2A) and males (Figure 2B).

### 3.4. Aquaporin Expression in Male and Female Reproductive Tissues and Cancers

Aquaporins in the reproductive systems of males are involved in formation of seminiferous fluid, sperm production, and motility [90]. In females, AQPs are vital for ovulation, vaginal lubrication, maturation of follicles, and maintenance of fluid homeostasis in the lumen of uterus [91]. AQPs 1, -2, -3, and -5 in the vagina are suggested to serve roles in tissue surface hydration [92], and in the uterus are proposed to maintain fluid homeostasis during peri-implantation and pregnancy [93,94]. Levels of AQP expression are dependent in part on hormonal signaling [93,95], controlled via steroid responsive promoters [96,97,98,99].

The classes of AQP channels that are associated with cancer severity and progression differ depending on the cancer type [8]. Differential aquaporin expression is evident in comparisons of prostate, breast, endometrial, ovarian, and cervical cancers [78,100,101,102,103]. For example, AQP1 is overexpressed in breast cancer [104] and colorectal cancer [105], whereas AQP3 is upregulated in prostate and breast cancers [102,106].

## 4. Estrogen-Dependent Tumors

Estrogen hormone is produced during reproductive years predominantly by ovaries in females and adrenal glands in males [107] to regulate physiological cell growth, and in females the development of sexual characteristics such as breasts, ovaries, and endometrium by activating estrogen receptors [108]. While the ovary is the main organ for production of estrogen, other tissues such as bone, brain, adipose tissue, and blood vessels use local production of estrogen by an aromatase enzyme to control bone mineral density, cholesterol metabolism, and cardio-protection [109,110].

Estrogen receptor (ER) protein types alpha and beta interact with DNA sequences known as estrogen-responsive elements (EREs). The receptors have a high affinity for estradiol, which is effective at nanomolar concentrations [111]. Gene expression controlled by EREs is essential for normal development but can be co-opted for pathological processes, as in estrogen-dependent tumors (Figure 3). ERα and ERβ receptor activities are involved in the progression of cancers in breast, endometrium, ovaries, and prostate gland [108]. Of the aquaporins present in breast, endometrial, and ovarian cancers, the most highly expressed subtypes are AQPs-1, -2, -3, and -5, all of which are sensitive to ERE signaling [96,99,112].

In females, the status of estrogen-dependent tumors is influenced by hormonal changes during the menstrual cycle [113,114,115]. In the follicular or proliferative phase of the menstrual cycle, estrogen is produced in high amounts, peaking around the time of ovulation, after which it declines [116]. Progesterone dominates in the post-ovulatory or luteal phase. Disturbance of the balance between estrogen and progesterone by secondary factors such as obesity, oral contraceptives, or polycystic ovarian disease can result in high levels of estrogen during the luteal phase, leading to uncontrolled proliferation of endometrium, breast and ovaries, resulting in hyperplasia, cytologic atypia, and cancer [117]. Anti-estrogen strategies have been a mainstay of breast cancer treatment for more than a century, since the majority of breast cancers are ER-positive [109]. Upregulation of AQP5 expression in uterus in response to estrogen [118] activates P13/AKT pathway, leading to cellular proliferation and increased invasiveness [119].

In males, estrogen is produced in testes, adrenal glands, and adipose tissue for maintaining reproductive function and libido [120]. Factors such as mutations in the gene encoding the P450 enzyme used for local estrogen synthesis, the use of certain drugs, and a positive family history influence the risk of development of estrogen-dependent tumors in males. Estrogen-dependent activation of the MEK-Erk1/2 (MAPK) pathway acts via ERα to promote cell proliferation [121]. Upregulation of AQP3 in prostate cancer cells contributes to tumor invasiveness by stimulating MEK-Erk1/2 (MAPK) pathway, and knockdown of AQP3 reduced tumor growth by blocking the same pathway [122], suggesting inhibition of AQP3 is a target of interest for developing potential treatments of prostate cancer.

Four types of estrogen-dependent tumors are summarized below: (i) breast, (ii) endometrial, (iii) ovarian, and (iv) prostate cancers.

### 4.1. Breast Cancer

Breast cancer is the most common cancer among females and the second most common cause of death worldwide [123]. Breast cancers are divided into three main groups on the basis of cellular markers: (1) estrogen receptor (ER)- or progesterone receptor (PR)-positive; (2) human epidermal growth factor receptor 2 (HER2)-positive with or without ER and PR positivity; and (3) triple-negative breast cancer (TNBC) defined by the absence of ER, PR, and HER2 expression [124]. Estrogen and progesterone receptor-positive tumors account for 70% of invasive breast cancers while TNBC constitutes 10% of invasive breast cancers [124]. Treatment options depend on metastatic spread and hormone receptor status. Metastatic breast cancer cells undergo an epithelial–mesenchymal transition that increases invasion of extracellular matrix and surrounding blood and lymph vessels, augmenting cell migration [125]. Breast cancers have a propensity to metastasize to bone (50–65%), lung (17%), brain (16%), and liver (6%), while metastases to other organs such as spleen, kidney, or uterus are relatively rare [126]. The incidence of breast cancer is high in reproductive years (18–50) and decreases after menopause; estradiol blood level shows a positive correlation with risk of breast cancer in females [127]. Early menarche, late menopause, nulliparity, use of combined oral contraceptives, and obesity constitute major risk factors for breast cancer [128].

Breast cancer tissues showed upregulation of AQP1, -3, and -5 at transcript and protein levels as compared to normal breast tissue, at levels dependent on cancer subtype and stage [100,104,129]. AQP1 levels in clinical cases positively correlated with histological grade, tumor size, lymph node metastasis, and distant metastasis; high AQP1 expression was associated with poor prognosis, increased recurrence, and a higher death rate within 5 years as compared to patients with low AQP1 expression in breast cancers [104]. After implantation in target organs, breast cancer metastatic tumors induce formation of new blood vessels [125] in response to both hypoxia-induced-factor-1 (HIF 1) and estrogen-activated AQP1 expression [99]. Tumor growth, VEGF signaling levels, vessel density, and lung metastases were reduced in AQP1 null mice compared to wild type [130]. Targeting AQP1 is emerging as a strategy of interest to reduce angiogenesis and tumor growth across a wide range of cancer types. 

Both increased and decreased levels of AQP3 expression have been suggested to have beneficial effects in breast cancer. Knockdown of AQP3 reduced motility in breast cancer cell lines in response to fibroblast growth factor [106,131], suggesting inhibition of AQP3 might control cancer cell migration. In contrast, an analysis of biopsy samples from HER2-positive early breast cancer patients showed that high levels of AQP3 protein correlated with longer periods of disease-free survival [100], pointing to a protective influence in vivo. Increased expression of AQP3 enhanced the effectiveness of the chemotherapeutic agent 5-fluorouracil, whereas downregulation of AQP3 reversed the cell cycle arrest [132], lowering chemotherapy effectiveness. Data for AQP5 thus far suggest a negative contribution. Combined levels of AQP3 and AQP5 protein in biopsy samples from TNBC patients were positively associated with tumor size, lymph node metastasis, and likelihood of relapse [133], indicating increased aquaporin expression was linked to poorer survival. Knockdown of AQP5 by short hairpin RNA reduced proliferation and migration [134,135]. Opposing conclusions on the role of AQPs in breast cancer likely reflect the complexity of the system, highlighted by the diversity of breast cancer subtypes, and differences in the experimental models used.

### 4.2. Endometrial Cancer

Endometrial cancer (EC) is the most common estrogen-dependent gynecological malignancy and the fifth or sixth most common cancer overall among females. In 2012, more than 300,000 women worldwide were diagnosed with EC [136]. Abnormal vaginal bleeding is an important diagnostic indicator, and treatment for EC in the non-invasive stage (stage I) is typically surgical, involving hysterectomy and bilateral salpingo-oophorectomy, with or without lymphadenectomy depending on the involvement of pelvic lymph nodes [137]. In biopsies from patients with different stage EC cancers, AQP3 protein expression was correlated with histological grade [138]. Regulated by an ERE promoter, increased expression of the AQP2 gene after estrogen treatment increased motility in EC cells [96]. Knockdown of AQP2 or AQP5 in EC reduced growth and invasiveness [112]. AQP1 in vascular endothelial cells also showed a positive correlation with tumor growth, histologic grade, and extra-uterine metastases [78], suggesting a parallel role for angiogenesis in aiding tumor growth and spread. Effective treatments are likely to require combined targeting of the AQPs involved in complementary processes, in the cancer cells directly and in the surrounding tissue environment.

### 4.3. Ovarian Cancer

Ovarian cancer is the sixth most common cancer overall and the fourth most common gynecological cancer worldwide [139]. The risk of occurrence in women of ovarian cancer is lowered by prolonged states of anovulation such as parity, reduced with first pregnancy and progressively lowered with subsequent full term births [140]. However, no association had been found between ovarian cancer and reproductive factors such as age of menarche, age of menopause, or breastfeeding [140]. Women usually present with nonspecific symptoms such as abdominal pain and distention, which makes diagnosis of the cancer difficult; retrospective analyses showed most cases were misdiagnosed as a gastrointestinal pathology such as irritable bowel syndrome [141]. Because diagnoses are often delayed until advanced stages, poor outcomes persist despite aggressive management [141].

AQPs 1, 2, 3, and 4 are expressed in granulosa and theca cells of non-cancerous ovarian tissues [142]. Increased expression of AQPs 1, 3, 5, and 7 has been found in borderline and malignant ovarian tumors [143]. AQP1 upregulation at transcript and protein levels in ovarian cancer cells was associated with increased cancer growth, while deletion of AQP1 reduced growth, migration, and invasion [144]. Malignant ascites [145], the most common complication of ovarian cancer, could involve AQP1 in the pathological accumulation of fluids out of blood vessels into the peritoneal cavity, secondary to increased permeability of capillaries and upregulation of VEGF [146].

Increased AQP3 protein in ovarian cancer cells was associated with EGF-stimulated growth and migration, blocked by the natural product curcumin [147]. AQP5 levels in ovarian cancer similarly have been correlated with tumor growth, as well as lymph node metastasis and volume of malignant ascites [148], suggesting AQP5 could be a prognostic factor [149]. AQP5 knockdown impaired growth and migration of epithelial ovarian cancer cells [150] and sensitivity to chemotherapy [151]. In contrast, the downregulation of AQP6 and 9, and stable levels of AQP8 in ovarian cancer [152], highlight the idea that the complex control systems for AQPs will require knowledge of subcellular localization, regulation by signaling, and all permeant solutes to fully explain mechanisms of action.

### 4.4. Prostate Cancer

Prostate cancer (PC) is the second most common cancer among men worldwide [153]. The risk factors include positive family history and prolonged exposure to testosterone and estrogen agonists [154]. Molecular screening has shown that a striking array of AQPs (1, 3, 5, 6, 7, 8, 9, 10, and 11) are expressed at protein and transcript levels in PC cell lines, in human benign prostatic hypertrophy, and in normal prostate tissues [102]. Low levels of expression of AQP3 and AQP9 were associated with more malignant PC tumors as compared to well differentiated PC and normal prostate tissues [102]. The potential benefit of high AQP3 levels in vivo contrasted with results from PC cell lines in which siRNA knockdown of AQP3 reduced migration and invasion [122]. Decreased expression of AQP3 in PC cells increased tumor sensitivity to cryotherapy, a method used to selectively destroy PC tumors while preserving vital structures such as bladder and bowel [155]. AQP9 gene silencing induced cell apoptosis and reduced migration and invasiveness of PC cells [156]. The aquaglyceroporin classes AQPs 3 and 9 appear to be interesting targets for intervention in prostate cancer progression, but mechanisms of action remain to be fully defined.

## 5. Non-Estrogen Dependent Tumors:

Tumors traditionally classified as non-estrogen-dependent (testicular and cervical) are characterized by a lack of ERs, yet, counterintuitively, effects of estrogen in the progression of these cancers have been noted. Actions of estrogen in initiation and progression of tumors previously designated as non-estrogen-dependent [157,158,159,160] suggests the hormone affects other cells or the tumor cell environment in testicular and cervical tissues to stimulate cancer progression, despite the absence of ERs on the cancer cells themselves. Properties of tumors classically described as estrogen-independent are summarized below for testicular and cervical cancers.

### 5.1. Testicular Cancer

Testicular cancer is the most common reproductive cancer among young men (15–44 years) in developed countries, accounting for 1% of male cancers worldwide. The risk of testicular cancer has been linked to exposure to environmental toxins in the maternal uterus [161], as well as other factors including cryptorchidism (undescended testes), male infertility, AIDS, low birth weight, and pre-mature birth [162]. A patient may present with painful testicular mass; scrotal swelling; or, more rarely, with metastatic features such as headache, low backache, or retroperitoneal lymphadenopathy [163]. Sites of metastasis are most commonly the local and distant lymph nodes, rete testis, and scrotum, and more rarely to the lungs and brain in late stages. An online database search for published work on AQPs associated with testicular cancer yielded no hits (PubMed, 12 Dec 2020), identifying a gap in knowledge in the field.

### 5.2. Cervical Cancer

Cervical cancer is the third most common cancer worldwide and the most frequent gynecological cancer among females [164]. The incidence of cervical cancer in developed countries has declined in recent years as a result of regular screening and early detection programs in accord with World Health Organization guidelines. Primary risk factors for cervical cancer are human papilloma virus (HPV) infection superimposed with HIV, and additional factors include an early age of starting sexual intercourse, multiple sexual partners, and unprotected sex [165]. Sites of metastasis typically involve the vagina, kidneys, and pelvic lymph nodes, and rarely more distant organs, treated by surgical resection in non-advanced cases and radiotherapy and/or chemotherapy in advances cases involving metastasis [165]. AQPs 1, 3, 5, and 8 are upregulated at both the transcript and protein levels in cervical intraepithelial neoplasia and cervical cancers as compared with normal tissue [166,167,168,169]. Increased AQP1 expression has been suggested to facilitate the progression from cervical intraepithelial neoplasia to cancer [101]. Increased AQP5 expression appears to be associated with lymph node involvement and poor prognosis in cervical cancer [167]. Expression, localization, interactions, and roles of aquaporins in cervical cancer remain an interesting area for further study.

## 6. Pharmacological Modulators of Aquaporins

An expanding array of pharmacological modulators of aquaporins include methylurea compounds [170], organic metal compounds [171,172,173], bumetanide derivatives and other arylsulfonamides [71,174,175,176], tetraethylammonium [177,178,179], and alternative medicinal constituents [180,181]. The diuretic acetazolamide, an antagonist of carbonic anhydrase approved for treatment of an array of diseases including heart failure [182], has been proposed to reduce AQP1 activity [183]. Acetazolamide reduced angiogenesis, tumor growth, and tumor invasion in the Lewis lung carcinoma mouse model [184], effects consistent with possible AQP1 downregulation [183,185], although other contributing mechanisms of action need to be considered [186].

A synthetic series of bumetanide (AqB) and furosemide (AqF) derivatives includes AQP1 inhibitors and activators [72,175]. AqB013, which blocks monomeric water pores, and AqB011, which blocks the AQP1 central pore, individually reduced invasion and migration in colon cancer cell lines, and together showed a synergist increase in anti-migration efficacy [71,176,187], suggesting AQP1 enhancement of cell motility depends on both water and ion channel functions. AqB013 induced apoptosis and inhibited tube formation in murine endothelial and human umbilical vascular endothelial cell lines, suggesting applications in reducing cancer-related angiogenesis [50,187].

Natural medicinal extracts are promising sources of AQP modulatory agents. Bacopasides I and II extracted from the Indian medicinal herb water hyssop (*Bacopa monnieri*) differentially inhibited AQP1-mediated water and ion fluxes and reduced motility, invasiveness, and lamellipodial formation of HT29 colon cells [72,180]. Inhibition of AQP1 intrasubunit pores by Bacopaside II impaired stress-induced H_2_O_2_ influx, protecting against hypertrophic heart disease in mouse and human cardiac models [54]. A Chinese herbal medicinal extract, compound Kushing injection, reduced growth and invasiveness of colon cancer, breast cancer, and glioblastoma cell lines and altered the expression of genes including AQPs [188]; however, specific actions on AQP functions remain to be defined. Ginsenoside was found to inhibit migration of prostate cancer cells in vitro by inhibiting AQP1-dependent pathways [189], while curcumin was found to inhibit AQP3-mediated cancer cell migration in human ovarian cancer cells [147]. The Chinese herbal medicine Fuling has been shown in a meta-analysis of clinical trials to decrease endometriosis recurrence, improve pregnancy outcomes, and relieve symptoms compared to standard therapy alone [190]. Evaluation of possible anti-AQP activities of components isolated from medicinal herb extracts merits further research. It is likely that many AQP modulatory agents await discovery. Ultimately, a comprehensive database would allow matching of AQP dependent pathways in cancer growth and metastasis with relevant pharmacological inhibitors for customized anti-cancer combinations.

### Aquaporins as Therapeutic Targets to Improve Cancer Outcomes

Cancer treatment options involve surgery, chemotherapy, and radiotherapy, which can be used in combination and augmented with anti-angiogenic and check point inhibitor agents. Despite progressive optimization of these tools to achieve the best clinical outcomes possible, each treatment modality is limited by adverse outcomes that impose mental and physical stress; psychological trauma; organ and tissue damage; and side effects such as intractable vomiting, diarrhea, bone marrow suppression, and sepsis. New options and adjunct therapies to augment cancer treatment are keenly needed. One avenue being explored for advancing cancer treatment strategies is targeting aquaporins. AQP knockout and overexpression models, genomic and proteomic analyses, and pharmacological inhibitors have provided lines of evidence suggesting potential value for pharmacological targeting of AQPs in controlling tumor growth and invasion (Figure 4). Diverse subtypes, patterns of expression, and functional roles in cancers offer an intriguing future for potentially high resolution control of specific aspects of pathology and cancer progress.

## 7. Conclusions

In normal physiology, AQPs serve as essential modulators of fluid transport and homeostasis in multiple organs and tissues. In pathological cancer conditions, aquaporins are implicated in the growth, migration, invasion, and angiogenesis, contributing to cancer progression and the life-threatening process of metastasis. Basic research and pre-clinical and clinical studies have demonstrated that the expression of certain aquaporins, notably AQP1 and AQP5, are increased in cancerous tissues compared to normal tissues, and correspond to level of malignancy. Others including the aquaglyceroporins can show an inverse association, with decreased levels of expression appearing to link to poor outcomes. Ongoing research will continue to elucidate properties and mechanisms of regulation of signaling pathways mediated by AQPs in tumorigenesis. Pharmacological blockers of aquaporin channels are being viewed as promising tools for improving cancer treatment, particularly in combinations that could synergistically slow the parallel processes of proliferation, angiogenesis, and invasiveness that subserve cancer progression (Figure 4) and render cancer therapy such a difficult healthcare challenge. Expanding research on pharmacological modulators of aquaporins is anticipated to hold breakthrough opportunities to improve outcomes for reproductive cancers, other cancer types, and more broadly for AQP-related diseases.

## Figures and Tables

**Figure 1 cells-10-00215-f001:**
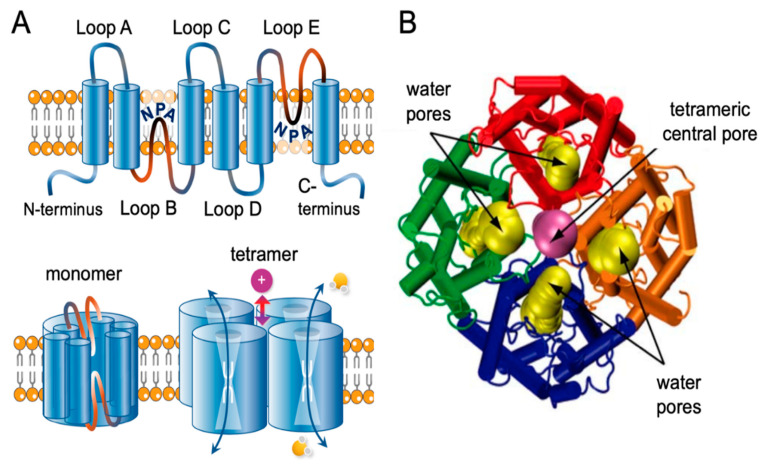
Transmembrane topology and structural organization of an aquaporin. (**A**) Each monomer consists of six transmembrane helices connected by loops A to E. Loops B and E typically carry signature NPA (Asn-Pro-Ala) motifs that fold together within each subunit to form a water pore. Four monomers form a tetramer (the functional channel). A subset of aquaporins (AQPs) use the central pore as a gated ion channel. (**B**) Molecular dynamic simulation view of the permeation properties of AQP1 depicted with space-filling models of pathways for water flow (yellow) through the intrasubunit pores, and monovalent cation (e.g., Na^+^) current (violet) through the central tetrameric pore, reproduced with permission from Elsevier, license #4967400727825 [66].

**Figure 2 cells-10-00215-f002:**
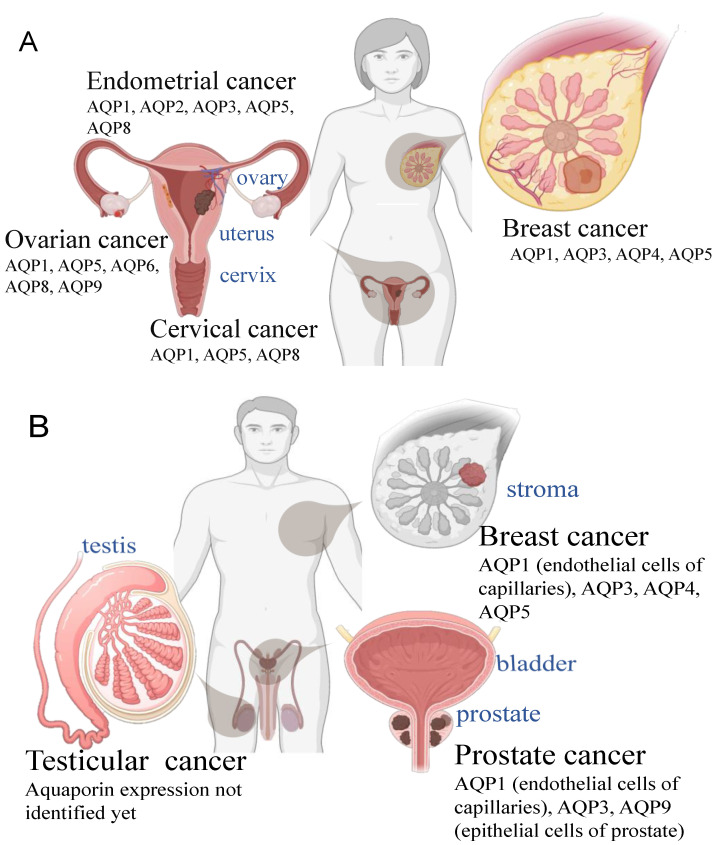
Illustrations of classes of aquaporins known to be expressed in breast and reproductive cancers, and general anatomical patterns of localization in human (**A**) female and (**B**) male bodies.

**Figure 3 cells-10-00215-f003:**
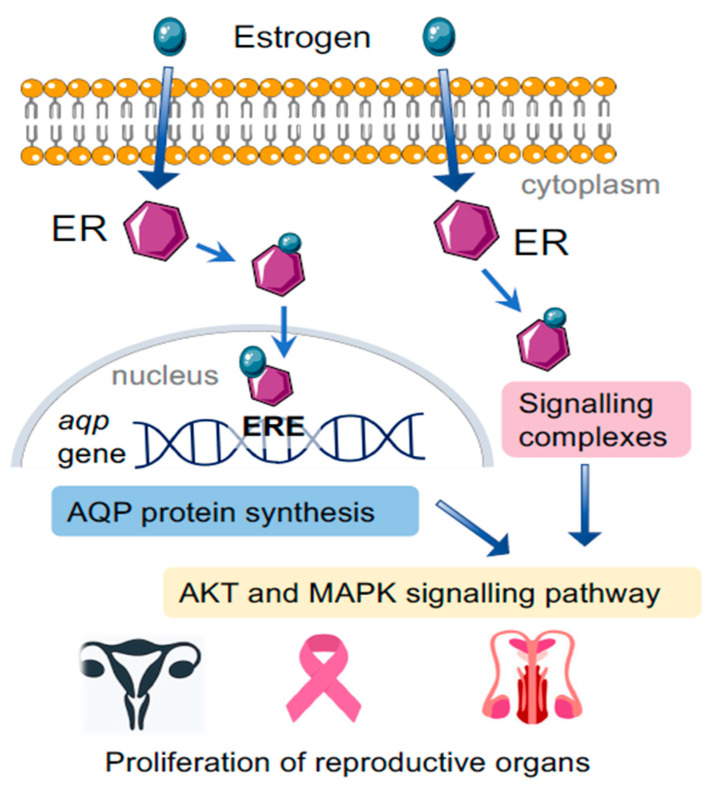
The intracellular signaling network activated by estrogen regulates gene expression by binding to DNA estrogen-responsive elements (EREs) for proteins including aquaporins, and activates downstream signaling pathways (P13/AKT and MEK-Erk1/2 (MAPK) pathway), promoting cell proliferation and invasiveness.

**Figure 4 cells-10-00215-f004:**
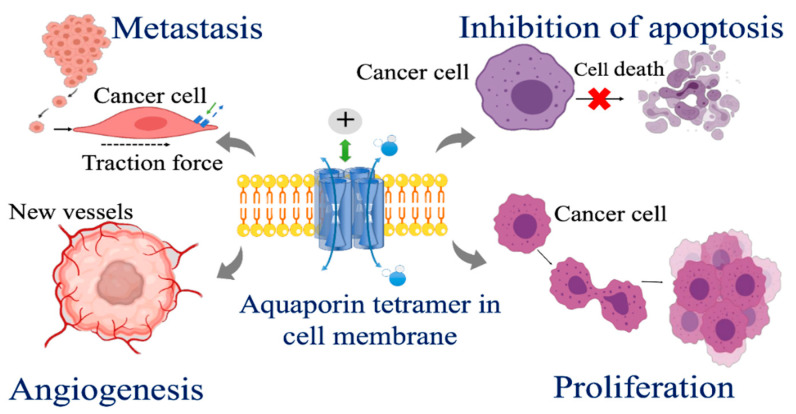
Diagram of the diverse areas of involvement of aquaporins in cancer growth and invasion. Facilitated transport of solutes and signaling molecules across cell membranes via aquaporin channels is linked to clinically important processes of cancer metastasis, angiogenesis, proliferation, and resistance to apoptosis.

**Table 1 cells-10-00215-t001:** Classes of aquaporins in higher mammals, prior alternative names, and references.

Aquaporin Class	Reference
AQP0 *(lens MIP)*	Gorin et al., 1984 [13]
AQP1 *(CHIP28)*	Preston et al., 1992. [5]
AQP2 *(WCH-CD)*	Fushimi et al., 1993. [14]
AQP3 *(GIL)*	Echevarria et al., 1994. [15] Ishibashi et al., 1994. [16]
AQP4 *(MIWC)*	Hasegawa et al., 1994. [17]
AQP5	Raina et al., 1995. [18]
AQP6 *(hKID)*	Ma et al., 1996. [19]
AQP7	Ishibashi et al., 1997. [20]
AQP8	Ishibashi et al., 1997. [21]
AQP9	Tsukaguchi et al., 1998. [22] Ishibashi et al., 1998. [23]
AQP10	Hatakeyama et al., 2001. [24] Ishibashi et al, 2002. [25]
AQP11 *(AQPX1)*	Yakata et al., 2007. [26]
AQP12A, AQP12B *(AQPX2)*	Itoh et al., 2005. [27]

## Data Availability

Not applicable.

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
