# Peer review of "Targeting Aquaporins in Novel Therapies for Male and Female Breast and Reproductive Cancers"

_cells, 2021, doi:10.3390/cells10020215_

Round 1
Reviewer 1 Report
Aquaporin proteins in the 2000s had been studied, focusing on the identification of permeating substances and their structural science in each subtype. After that, most research has shifted to investigating the pathophysiological significance of each subtype. This review is a timely paper detailing each aquaporin subtype's role in male and female reproductive cancers. Furthermore, the potential of molecules that regulate aquaporin proteins as anticancer agents is also properly introduced. From the above, this paper is considered to be very useful for medical professionals involved in the oncology area, especially reproductive cancers.
This review article is very interesting because it explains a new field of aquaporin research, especially suggesting new treatments for male and female reproductive cancers. Please let me only point out the following minor points.
Minor comments
1. Page 9, 2nd para, line 12: “such s” might be “such as”?
2. There is a capital letter in the middle of the text other than the sentence's beginning.
Reviewer 2 Report
This is an interesting review that focuses on Targeting Aquaporins in Novel Therapies for Male and Female Reproductive Cancers but it is not acceptable in the present form.
Comments
The study is complessively well written and figures are clear and representative. However I would suggest several changes to make it more more readily understandable.
1.The title sould be modified, and the word reproductive should be changed. In fact in the main text breast cancer is discussed, however breast is not a reproductive organ.
-
In the Abstract section major intrinsic proteins 10 (MIPs) are cited as acronym at the first citation but acquaporin not, please correct.
-
In the Introduction section Aquaporins (AQPs) are cited as acronym at the first citation but major intrinsic proteins not, please correct.
-
At line 99 I read: ^Current cancer treatments are focused on inhibiting cell proliferation and arresting 99 cell cycle, but tools for controlling metastasis remain elusive^.
This is only in part realy in that stromal microenvironment is also a target of antiangiogenic drugs and check points inhibitors.
The above sentence should be improved and expanded.
-
At line 117 VEGF shoud be detailed in full and then in acronym as first citation, please correct.
-
Again at line 399 the sentence ^Cancer treatment options involve surgery, chemotherapy or radiotherapy alone or in combination^ should be improved and expanded as above suggested.
-
A table grouping the main acquaporins classes should be included in the main text.
-
Finally there are too many citations (176)Nin the References section and it is hard follow all of them. I suggest to insert only the more recent and more important of them.
Round 2
Reviewer 2 Report
The manuscript is now suitable for publication